# Enhancing COVID-19 Knowledge among Nursing Students: A Quantitative Study of a Digital Serious Game Intervention

**DOI:** 10.3390/healthcare12111066

**Published:** 2024-05-23

**Authors:** Hannah McConnell, Debbie Duncan, Patrick Stark, Tara Anderson, James McMahon, Laura Creighton, Stephanie Craig, Gillian Carter, Alison Smart, Abdulelah Alanazi, Gary Mitchell

**Affiliations:** 1School of Nursing and Midwifery, Queen’s University Belfast, Belfast BT7 1NN, UK; h.mcconnell@qub.ac.uk (H.M.); d.duncan@qub.ac.uk (D.D.); p.stark@qub.ac.uk (P.S.); tanderson@qub.ac.uk (T.A.); j.mcmahon@qub.ac.uk (J.M.); laura.creighton@qub.ac.uk (L.C.); scraig22@qub.ac.uk (S.C.); g.carter@qub.ac.uk (G.C.); a.smart@qub.ac.uk (A.S.); aalanazi05@qub.ac.uk (A.A.); 2Department of Nursing, Faculty of Applied Medical Sciences, The University of Bisha, Bisha 67714, Saudi Arabia

**Keywords:** COVID-19, serious games, gamification, nursing education, healthcare education, education, health care, vaccine hesitancy

## Abstract

Background: This study aimed to evaluate the effectiveness of a digital serious game intervention about COVID-19 on pre-registration nursing student knowledge. Method: This study included 282 nursing students from a university in Northern Ireland, with 210 students providing comparable pre-test and post-test results. The ‘serious game’ aimed to debunk common COVID-19 myths and provide accurate information about the virus. Participants completed a 25-item questionnaire before and after engaging with the game, which included true/false items based on the World Health Organisation’s list of top COVID-19 myths. The data were analysed using paired *t*-tests to assess knowledge changes, and scores were calculated as percentages of correct answers. Results: A statistically significant improvement in COVID-19 knowledge was demonstrated among first-year nursing students who engaged with the serious game. The post-test scores (M = 92.68, SD = 13.59) were notably higher than the pre-test scores (M = 82.64, SD = 13.26), with a *p*-value less than 0.001. Conclusion: This research suggests that integrating serious games into undergraduate nursing education can effectively enhance COVID-19 knowledge. This approach is aligned with the evolving trend of employing technology and gamification in healthcare education.

## 1. Introduction

The COVID-19 pandemic has emerged as one of the most significant health crises in modern history, profoundly impacting societies worldwide [1,2,3,4]. Its far-reaching effects have spurred unprecedented shifts in human behaviour across diverse spheres [5,6]. In response to varying degrees of prevalence and severity, nations have implemented a spectrum of measures ranging from stringent adherence to health protocols and social distancing mandates to widescale lockdowns and travel restrictions, alongside the closure of venues facilitating mass gatherings [6,7]. These measures, while pivotal for curbing transmission, have reshaped daily routines and societal norms, highlighting the pandemic’s profound influence on global dynamics.

Clinical manifestations have been shown to include fever, cough, fatigue, hoarseness, and diarrhoea and a loss of smell, although there have also been identified asymptomatic carriers of the virus [6,8]. Transmission of the virus is through droplet spread, although there is emerging evidence that it may also be through the faecal–oral route, too [9]. People who are middle-aged or older are more susceptible to infection [10]. There are, however, high rates of morbidity and mortality in healthy, young adult patients [11,12]. Other risk factors are respiratory disease, smoking, obesity, and diabetes, although there are patients who have had it with no risk factors [13,14,15,16]. The estimated infection fatality risk (IFR) is thought to increase with repeated exposure to the virus [17].

It therefore remains important to educate pre-registration student nurses about COVID-19 during their training as it may have implications for them in the future. Health staff have experienced post-traumatic stress disorder due to their experiences of caring for patients with COVID-19 [18,19,20]. Qualified and pre-registration nurses have also died due to COVID-19 outbreaks around the world [21,22,23]. The pandemic continues to influence levels of anxiety, fear, and worry for the undergraduate nurse [24]. Despite these outcomes, studies have shown that the pandemic can positively re-enforce the aspiration for some people to become nurses [25,26].

Recent studies [27,28,29,30] have highlighted the potential of serious games in addressing COVID-19-related knowledge gaps and promoting behaviour change among various populations. For instance, games like Escape COVID-19 and Plague Inc. have been designed to teach infection prevention and control practices, contributing to increased knowledge and attitudes regarding COVID-19 [27,28]. Additionally, serious games like Point of Contact (PoC) have demonstrated success in improving young adults’ perceptions of preventive measures, fostering greater awareness and compliance with guidelines [29]. Furthermore, serious games have proven effective in healthcare settings, particularly among frontline workers. Studies have shown that games like Escape COVID-19 have successfully influenced healthcare personnel to adopt COVID-19-safe practices, indicating their potential to enhance IPC behaviours on a national scale [27]. Similarly, serious games like COVIDgame have been instrumental in medical education, providing an innovative and engaging platform for students to acquire and retain essential knowledge about COVID-19 [30].

These findings are not surprising. Digital gaming or ‘gamification’ is becoming an increasingly common way to educate employees in both the business and health sectors [31]. The use of ‘serious games’ continues to gain prevalence in healthcare and education settings and is renowned as an effective and innovative educational tool [32,33,34,35,36]. Indeed, because of COVID-19 and subsequent advancement in digitisation, the integration of digital technologies and gamification is arguably more relevant than ever before [37]. In contrast to games that are played for entertainment purposes, the aim of a serious game is to meet a specific educational need, with content delivered in an interactive format [33]. Through engagement with the serious game, players develop their knowledge and understanding around a topic. In the context of health care, serious games have been found to be beneficial to students and registered professionals, providing a unique format to challenge people’s thinking in a creative way [33,34,35]. 

While serious games show promise in addressing COVID-19 challenges, there are notable limitations [27,28,29,30]. These include uncertainties regarding their long-term effectiveness in changing behaviours and the variability in their quality and design. Self-reported measures used in studies may introduce bias, and scalability and accessibility issues remain significant challenges, particularly in low-resource settings. Additionally, the rapid evolution of the pandemic may render some serious games outdated. Addressing these concerns is crucial for maximising the utility of serious games in combating COVID-19. To our knowledge, there has been no serious game that was co-designed, digital, asynchronous, and designed for solo play. The intervention used in this study focuses on debunking common myths and misconceptions about COVID-19, which remain a problem even after 5 years since the first case of COVID-19, highlighting its importance. Furthermore, its digital and asynchronous format allows for easy updates, ensuring it always reflects the most recent evidence and guidelines. The aim of this study was therefore to evaluate the effectiveness of a ‘serious game’ around COVID-19 awareness on first-year undergraduate nursing students. The objective of this study was to determine if playing a COVID-19 awareness game improved year-one nursing students’ knowledge about COVID-19. 

## 2. Materials and Methods

### 2.1. Ethics

This study was approved by the Faculty of Medicine, Health, and Life Sciences Research Ethics Committee (MHLS20_107). Participants were not required to provide written or verbal consent for participation in the study. However, they were explicitly informed that their participation in any of the questionnaires was entirely voluntary. Consent was implied when participants accessed the surveys and chose to complete them.

### 2.2. Design, Setting, and Population

This study took place in one university in Northern Ireland between 4 January 2021 and 28 January 2022. All eligible participants (n = 412) were undertaking their first year of a BSc honours degree in professional nursing. Participants were sampled from all four fields of nursing (adult, mental health, learning disabilities, and children). All eligible participants were provided with access to the ‘serious game’ for a total of 30 days. Access to the serious game was affiliated with a year-one module, but playing the game and study participation was optional. Students could also choose to participate in the game and not participate in the research study. In total, 282 nursing students, a response rate of 68.45%, participated and completed a 25-item questionnaire comprised of true/false questions to determine if engagement with the game contributed to knowledge uptake on COVID-19 (Appendix A). The Transparent Reporting of Evaluations with Nonrandomised Designs (TREND) reporting guideline informed the reporting of this intervention evaluation and is available from the corresponding author on reasonable request.

### 2.3. Intervention

The game, known as ‘Coronavirus—know the facts’, was developed by Focus Games Ltd. in 2020 [38]. The purpose of the game was to promote knowledge about COVID-19 and to demystify some of the myths and common misconceptions surrounding the virus. Due to the rapid transmission of COVID-19 and the evolving guidance, there were heightened instances of the spread of misinformation through media forums [39,40]. 

The COVID-19 game is an HTML5 application that can be accessed freely on computer, tablet, or mobile devices. The game is relatively short, taking less than five minutes to play, with players answering true/false questions on facts and common myths associated with COVID-19. Common myths include “Coronavirus only affects older people”, “The flu vaccine will protect me against coronavirus”, and “I am pregnant so I’m more at risk of catching coronavirus”. Questions are randomly generated from a pre-existing question bank, and the game follows a ‘honeycomb path’; if a player answers a question correctly, they are presented with several hexagons to choose from and determine where to go next. The aim is to build a path that avoids the coronavirus particles on the screen and to answer as many questions as possible correctly. With each answer, the player will receive feedback and information related to their chosen question. As questions are randomly generated, players can have multiple attempts. The overall objective of the serious game is to improve awareness of the virus and to debunk the myths associated with COVID-19. 

### 2.4. Consent and Recruitment

All students, totalling 412, were contacted via email and provided with comprehensive information about the serious game by a person not directly involved in the project. This step ensured that the dissemination of information was impartial and transparent. The provided information sheet outlined the purpose of the research, clarified the consent process, and detailed how participant data would be anonymised to protect privacy and confidentiality. Furthermore, to maintain transparency, the project details and contact information of the research team were readily accessible on the module page where students could access the COVID-19 game.

Students who opted to participate in the evaluation were granted access to the pre- and post-questionnaire links on their individual module homepage. It is important to note that completion of the questionnaires was voluntary; students were not obligated to fill them out to access the game. This approach aimed to uphold the principles of voluntary participation and informed consent while allowing students the flexibility to engage with the game and research activities at their discretion. By providing clear information and accessibility to both the serious game and research materials, the study sought to ensure transparency, respect participants’ autonomy, and promote ethical conduct throughout the research process.

### 2.5. Data Collection

A pre-test, post-test design was used for this study. Students who participated in the study completed a questionnaire immediately before they played the game and immediately after playing. The questionnaire was designed by the research team and piloted with 15 second-year nursing students prior to administration. The questionnaire was written in English and consisted of 25 true or false items based on the World Health Organisation’s list of top COVID-19 myths (https://www.who.int/emergencies/diseases/novel-coronavirus-2019/advice-for-public/myth-busters, accessed on 22 December 2023). Importantly, none of the questionnaire items were explicitly presented in the serious game (e.g., the student did not receive an identical question and answer from the serious game that was on the post-questionnaire). A copy of the 25-item questionnaire can be viewed at Appendix A.

### 2.6. Data Analysis

The questionnaires were administered to participants before and after they played the serious game. This analysed using paired *t*-tests to determine if the ‘serious game’ increased nursing students’ knowledge and understanding of COVID-19. Scores were calculated as the percentage of correct answers for pre-test and post-test total scores and sub-scales in SPSS version 27. All analyses were conducted using the percentage-based scores. Descriptive statistics were calculated for pre-test and post-test scores. Distribution of pre-test scores was examined using a histogram to investigate any floor or ceiling effects or potential for regression to the mean when comparing with post-test scores. A paired *t*-test was conducted to examine the change from pre-test to post-test for total score. 

## 3. Results

In total, 282 nursing students participated in the pre-test, post-test, or both questionnaires. Primary analysis was possible for 210 cases out of a total of 282 participants for whom data was recorded at both pre-test and post-test. Participants were not able to be included in the primary analysis paired comparison if they were missing either the pre-test or post-test, but their data were included in the descriptive statistics. Missing data (see Table 1) occurred due to a participant not sitting either the pre-test or post-test or due to participants not supplying a correct identifier (name or student number) to allow their pre-test and post-test data to be matched, despite completing the survey.

### Primary Analysis Results

Descriptive statistics provided insights into the scores obtained by participants. The mean (M) represents the average score, while the standard deviation (SD) indicates the spread or variability of scores around the mean. In simpler terms, the mean tells us the typical score achieved, while the standard deviation informs us how much the scores deviate from this typical value.

The paired-samples *t*-test was used to compare the mean scores of the same group of participants before and after an intervention—in this case, the completion of pre-test and post-test questionnaires. It helps determine whether the observed differences between the two sets of scores are statistically significant or merely due to chance.

The *t*-test result (*t*(209) = 14.55, *p* < 0.001) consists of two components: the t-value and the *p*-value. The t-value (14.55) quantifies the size of the difference between the pre-test and post-test scores relative to the variability in the data. A larger t-value indicates a greater difference between the means. The *p*-value (<0.001) indicates the probability of observing such a difference purely by chance. In this case, the *p*-value is less than 0.001, suggesting that the observed difference is highly unlikely to occur by random chance alone. Therefore, the results of the *t*-test indicate a statistically significant difference between the pre-test and post-test scores, suggesting that the intervention had a significant impact on the participants’ scores. The distribution of pre-test scores (Figure 1) shows that there is an approximately normal distribution with no major positive or negative skew or evidence of ceiling or floor effects.

Descriptive statistics (Table 2) show that post-test scores (M = 92.68, SD = 13.59) were higher than pre-test scores (M = 82.64, SD = 13.26). The difference between these two scores was statistically significant as indicated by the paired-samples *t*-test, which gave a result of t (209) = 14.55, *p* < 0.001.

## 4. Discussion

The pandemic-driven shift to online education has highlighted the critical role of technology-mediated learning, widely acknowledged for its importance in ensuring educational equity and inclusivity [41,42,43]. Gamification, a strategy with enriched experiential learning, offers students dynamic and innovative avenues to explore various subjects, fostering deeper engagement and understanding [32,33,34,35,36,44,45]. This study’s findings illuminate how pre-registration nursing students’ interaction with the COVID-19 serious game enriched their understanding and awareness, empowering them to discern factual information from common misconceptions [46,47]. Despite the well-documented challenges posed by online learning, uptake and participation in this study was high.

Several studies have delved into the efficacy of serious games as an innovative educational tool, demonstrating their potential to enhance understanding of COVID-19 and foster safe infection prevention and control behaviours [27,28,29,30]. Building upon this existing body of literature, the present study provided a cohort of first-year undergraduate nursing students with access to a serious game designed to address COVID-19-related knowledge and awareness over a period of 30 days. Incorporating insights from prior research, the study utilised a 25-item questionnaire to assess students’ knowledge levels before and after engaging with the COVID-19 serious game. Analysis of pre- and post-test results revealed statistically significant changes in students’ knowledge levels following gameplay, aligning with findings from previous COVID-19 serious game studies [27,28,29,30,46,47]. This highlights the potential of serious games as effective educational interventions for enhancing nursing students’ understanding of critical public health issues such as COVID-19. Like similar serious game studies, the integration of serious games into nursing education therefore appears to offer an engaging and interactive learning experience and to promote active participation and knowledge retention among students [33,34]. By immersing themselves in gameplay, student learners can explore real-world scenarios and apply theoretical knowledge in practical contexts, thereby reinforcing the key concepts and skills essential for effective nursing practice.

While this study focused on information uptake relating to COVID-19, the results of this initial study could be of interest in the context of student nurse uptake of the COVID-19 vaccination. Vaccine hesitancy for previous vaccines amongst healthcare professional groups has been a challenge for several years, even though it is recognised as one of the most successful public health measures [48,49]. There have been several research studies that have examined healthcare professional views on COVID-19 vaccination [50,51]. The main theme around vaccine hesitancy seems to be related to the rapid developments of COVID-19 vaccines globally and concerns about the safety of such vaccines. Other research highlights concerns about potential side effects of the vaccine [52]. Considering the findings of this study, which demonstrated high engagement and knowledge acquisition through the use of a serious game in the context of COVID-19 education, educators and healthcare institutions may consider leveraging similar gamified approaches to address vaccine hesitancy among nursing students. Serious games could serve as a platform to provide accurate information about COVID-19 vaccines, dispel myths and misconceptions, and address concerns regarding safety and efficacy. Moreover, incorporating interactive elements such as decision-making scenarios related to vaccine administration and public health measures could help nursing students develop the skills and confidence necessary to advocate for vaccination both in clinical practice and within their communities.

### 4.1. Recommendations

Future research in the field of healthcare education and serious games should explore the subtle, long-term effects of integrating such interactive technologies into curricula, investigating how they continue to shape the knowledge and clinical decision-making abilities of nursing students as they progress through their education and enter the healthcare workforce. Additionally, the development and assessment of serious games for a broader spectrum of healthcare-related topics, beyond COVID-19, is essential, enabling the creation of engaging, interactive learning tools that cater to diverse healthcare subjects. Further exploration should include the expansion of serious games to support the continuous education and professional development of qualified nurses and other healthcare professionals, with a focus on enhancing clinical skills, maintaining up-to-date knowledge, and addressing evolving healthcare challenges.

The findings derived from the serious game hold promise for augmenting healthcare education, particularly within nursing. It is recommended that academic institutions consider the integration of serious games into their educational frameworks to complement conventional pedagogical methodologies [53]. Such integration stands to furnish students with immersive, experiential learning modalities, thereby enriching their comprehension of pivotal healthcare tenets, including the management of COVID-19. Furthermore, given the critical importance of vaccine acceptance in combating infectious diseases such as COVID-19, efforts could also be directed towards leveraging serious games as brief interventions to address myths associated with vaccine hesitancy among healthcare professional students and healthcare workers.

### 4.2. Strengths and Limitations

The study on the effectiveness of a serious game in enhancing COVID-19 awareness among first-year undergraduate nursing students possesses several notable strengths and limitations. One of its key strengths lies in its contribution to the evolving landscape of healthcare education, demonstrating the potential of serious games to enhance students’ knowledge and critical thinking skills on a crucial health topic, like COVID-19. The study’s design, employing a pre-test and post-test model, allowed for the assessment of knowledge improvement, offering valuable insights into the impact of the game on students. The inclusion of a large sample of first-year nursing students from various fields and its voluntary participation aspect reflected real-world conditions. However, the study does carry limitations, including a notable drop in the number of participants with comparable pre-test and post-test results, potentially affecting the generalisability of the findings. The absence of a control group or alternative intervention limits the ability to attribute knowledge gains solely to the serious game. Furthermore, the study primarily focused on first-year students, and its transferability to second- and third-year students remains uncertain. The use of self-reported questionnaires and knowledge assessments may introduce a degree of response bias and subjectivity, potentially impacting the robustness of the results. Despite the absence of formal validation procedures, the questionnaire’s grounding in authoritative sources such as the WHO, coupled with prior pilot testing, bolsters the confidence in the study’s methodology. Future research could explore additional validation methods, such as more extensive psychometric analyses and iterative piloting, to further strengthen the questionnaire’s reliability. Nevertheless, for the purposes of this study, the questionnaire’s alignment with WHO guidelines provided a solid basis for data collection and analysis.

Nevertheless, the study provides valuable foundational evidence regarding the educational potential of serious games in health care, offering opportunities for further research and development in this dynamic field.

## 5. Conclusions

This study demonstrates the efficacy of incorporating serious games into undergraduate nursing education to improve COVID-19 knowledge and debunk myths. Through the implementation of a pre-test and post-test design, significant improvements in participants’ knowledge levels were observed following engagement with the COVID-19 serious game. These findings highlight the potential of serious games as innovative educational interventions for promoting active learning and fostering the essential skills requisite for effective nursing practice. The study’s recommendations advocate for the integration of serious games into nursing curricula to complement traditional pedagogical approaches, providing students with immersive, experiential learning modalities. Furthermore, given the imperative of vaccine acceptance in combating infectious diseases like COVID-19, serious games may offer a promising avenue for addressing vaccine hesitancy among healthcare professional students and workers. 

While some limitations exist, this research underscores the potential for serious games to enhance students’ awareness and knowledge, ultimately contributing to their preparedness for safe clinical practice in the face of evolving health challenges.

## Figures and Tables

**Figure 1 healthcare-12-01066-f001:**
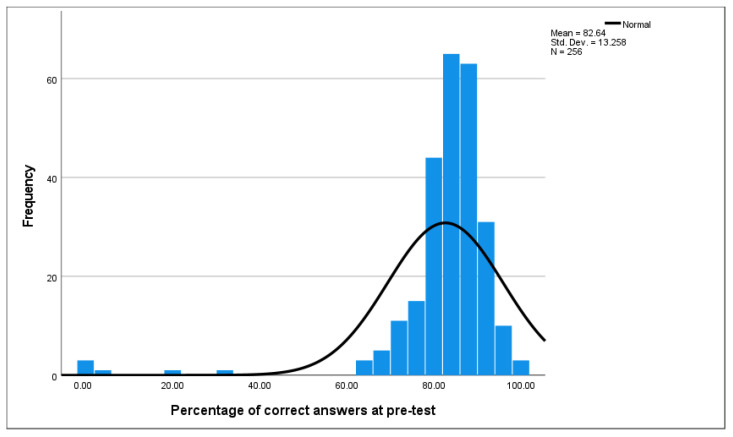
Pre-test scores.

**Table 1 healthcare-12-01066-t001:** Missing data.

Data	Missing n (%)
Pre-test data	26 (9.22%)
Post-test data	46 (16.31%)
Paired comparisons	72 (25.53%)

**Table 2 healthcare-12-01066-t002:** Descriptive statistics for pre-test and post-test total scores.

	N	Minimum	Maximum	Mean	Std. Deviation
Pre-test total % correct	256	0.00	100.00	82.64	13.26
Post-test total % correct	236	0.00	100.00	92.68	13.59
Valid N (listwise)	210				

## Data Availability

The datasets generated and/or analysed during the current study are available from the corresponding author upon reasonable request.

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
