# Peer review of "Enhancing COVID-19 Knowledge among Nursing Students: A Quantitative Study of a Digital Serious Game Intervention"

_healthcare, 2024, doi:10.3390/healthcare12111066_

Round 1
Reviewer 1 Report
Comments and Suggestions for Authors
Dear Authors,
Here are some suggestions:
1. The current title, "Evaluation of a serious game on nursing student awareness of 2 COVID-19: A pre-test post-test study," while informative, could be improved to better capture the essence of the research and its potential impact for a high-impact journal. Shift the focus from the evaluation aspect to the game itself. Highlight the game's purpose and potential benefits. Avoid indicating the research methodology: Including "pre-test post-test study" in the title can make it sound less impactful and potentially less suitable for a high-impact journal.
2. The provided background information about COVID-19 in 1st paragraph is currently outdated. While it accurately describes the origins and characteristics of the virus, it's important to focus on the present scenario and the ongoing need for awareness, particularly among students like nursing students.
3.Your study's focus on evaluating a serious game for improving COVID-19 awareness among nursing students is commendable. However, the introduction is general information about COVID-19 effect which is widely known. Since you are talking about game your introduction should: a. Briefly mention the existing literature on serious games or something similar in nursing education, particularly those focused on COVID-19 awareness.
b. Highlight inconsistencies or limitations: Identify any limitations, inconsistencies, or areas of controversy in the existing research.
c. Specify your contribution: Explain how your study addresses these gaps. Does your game focus on a specific aspect of COVID-19 knowledge not previously explored? Does it utilize a unique evaluation method or game design element?
4. Your manuscript includes statistical results, which is commendable. However, no brief explanation of the key statistical calculations employed to enhance reader comprehension.
Current Example:
Descriptive statistics (Table 2) show that post-test scores (M=92.68, SD=13.59) were 203 higher than pre-test scores (M=82.64, SD=13.26). The difference between these two scores 204 was statistically significant as indicated by the paired samples t-test which gave a result 205 of t (209) =14.55, p<0.001.
Briefly define the mean (M) and standard deviation (SD) for readers unfamiliar with these terms.
Explain the purpose of the paired-samples t-test in plain language (e.g., compares means from the same group before and after).
Decipher the t-test result (t(209) = 14.55, p < 0.001) by explaining what the t-value and p-value represent and how they indicate a statistically significant difference.
Do the same to other statistical calculation. By providing these clarifications, you can ensure that readers with varying statistical backgrounds understand the significance of your findings.
5. Your manuscript outlines two objectives:
a.To determine if playing a COVID-19 awareness game improved year one nursing students' knowledge about COVID-19.
b.To identify if nursing students were able to identify facts from myths in relation to the transmission of and protective measures required in relation to COVID-19.
However, the results section appears to focus solely on the first objective, analyzing the overall improvement in knowledge scores. The second objective, regarding identifying facts from myths, is not addressed.
Overall, your manuscript has the potential to be a valuable contribution to the field of nursing education. However, before it is ready for publication in a high-impact journal, I recommend addressing the points I specified.
Author Response
Reviewer Report 1
- The current title, "Evaluation of a serious game on nursing student awareness of 2 COVID-19: A pre-test post-test study," while informative, could be improved to better capture the essence of the research and its potential impact for a high-impact journal. Shift the focus from the evaluation aspect to the game itself. Highlight the game's purpose and potential benefits. Avoid indicating the research methodology: Including "pre-test post-test study" in the title can make it sound less impactful and potentially less suitable for a high-impact journal.
Thank you. We have now amended our title to: Enhancing COVID-19 Knowledge Among Nursing Students: A Quantitative Study of a Digital Serious Game Intervention
- The provided background information about COVID-19 in 1st paragraph is currently outdated. While it accurately describes the origins and characteristics of the virus, it's important to focus on the present scenario and the ongoing need for awareness, particularly among students like nursing students.
Thank you. We have revised paragraph one in accordance with your feedback. It now reads: “The COVID-19 pandemic has emerged as one of the most significant health crises in modern history, profoundly impacting societies worldwide [1-4]. Its far-reaching effects have spurred unprecedented shifts in human behaviour across diverse spheres [5-6]. In response to varying degrees of prevalence and severity, nations have implemented a spectrum of measures ranging from stringent adherence to health protocols and social distancing mandates to widescale lockdowns and travel restrictions, alongside the closure of venues facilitating mass gatherings [6-7]. These measures, while pivotal for curbing transmission, have reshaped daily routines and societal norms, highlighting the pandemic's profound influence on global dynamics”.
We also provide information on the importance of focusing on pre-registration nursing students in paragraph three.
3.Your study's focus on evaluating a serious game for improving COVID-19 awareness among nursing students is commendable. However, the introduction is general information about COVID-19 effect which is widely known. Since you are talking about game your introduction should: a. Briefly mention the existing literature on serious games or something similar in nursing education, particularly those focused on COVID-19 awareness.
Thank you for this helpful feedback. We have added the following paragraph in to provide a synthesis of recent evidence: “Recent studies have highlighted the potential of serious games in addressing COVID-19-related knowledge gaps and promoting behavior change among various populations. For instance, games like Escape COVID-19 and Plague Inc. have been designed to teach infection prevention and control practices, contributing to increased knowledge and attitudes regarding COVID-19 [27-28]. Additionally, serious games like Point of Contact (PoC) have demonstrated success in improving young adults' perceptions of preventive measures, fostering greater awareness and compliance with guidelines [29]. Furthermore, serious games have proven effective in healthcare settings, particularly among frontline workers. Studies have shown that games like Escape COVID-19 have successfully influenced healthcare personnel to adopt COVID-19-safe practices, indicating their potential to enhance IPC behaviours on a national scale [27]. Similarly, serious games like Covidgame have been instrumental in medical education, providing an innovative and engaging platform for students to acquire and retain essential knowledge about COVID-19 [30].”
- Highlight inconsistencies or limitations: Identify any limitations, inconsistencies, or areas of controversy in the existing research.
- Specify your contribution: Explain how your study addresses these gaps. Does your game focus on a specific aspect of COVID-19 knowledge not previously explored? Does it utilize a unique evaluation method or game design element?
Thank you. We have now added the following paragraph: “While serious games show promise in addressing COVID-19 challenges, there are notable limitations [27-30]. These include uncertainties regarding their long-term effectiveness in changing behaviors and the variability in their quality and design. Self-reported measures used in studies may introduce bias, and scalability and accessibility issues remain significant challenges, particularly in low-resource settings. Additionally, the rapid evolution of the pandemic may render some serious games outdated. Addressing these concerns is crucial for maximizing the utility of serious games in combating COVID-19. To our knowledge, there has been no serious game that is co-designed, digital, asynchronous, and designed for solo play. The intervention used in this study focuses on debunking common myths and misconceptions about COVID-19, which remains a problem even after 5 years from the first case of COVID-19, highlighting its importance. Furthermore, its digital and asynchronous format allows for easy updates, ensuring it always reflects the most recent evidence and guidelines”.
- Your manuscript includes statistical results, which is commendable. However, no brief explanation of the key statistical calculations employed to enhance reader comprehension.
Current Example:
Descriptive statistics (Table 2) show that post-test scores (M=92.68, SD=13.59) were 203 higher than pre-test scores (M=82.64, SD=13.26). The difference between these two scores 204 was statistically significant as indicated by the paired samples t-test which gave a result 205 of t (209) =14.55, p<0.001.
Briefly define the mean (M) and standard deviation (SD) for readers unfamiliar with these terms.
Explain the purpose of the paired-samples t-test in plain language (e.g., compares means from the same group before and after).
Decipher the t-test result (t(209) = 14.55, p < 0.001) by explaining what the t-value and p-value represent and how they indicate a statistically significant difference.
Do the same to other statistical calculation. By providing these clarifications, you can ensure that readers with varying statistical backgrounds understand the significance of your findings.
Thank you. We have rewritten the results for clarity: “The study involved 282 nursing students who participated in pre-test, post-test, or both questionnaires. Out of these, data from 210 participants were analysed for both pre-test and post-test. Missing data, totalling 25.53% for paired comparisons, occurred primarily because some participants did not complete either the pre-test or post-test. This exclusion criterion ensured that only complete data pairs were used for the primary analysis. Descriptive statistics provided insights into the scores obtained by participants. The mean (M) represents the average score, while the standard deviation (SD) indicates the spread or variability of scores around the mean. In simpler terms, the mean tells us the typical score achieved, while the standard deviation informs us how much the scores deviate from this typical value.
The paired-samples t-test was used to compare the mean scores of the same group of participants before and after an intervention, in this case, the completion of pre-test and post-test questionnaires. It helps determine whether the observed differences between the two sets of scores are statistically significant or merely due to chance.
The t-test result (t(209) = 14.55, p < 0.001) consists of two components: the t-value and the p-value. The t-value (14.55) quantifies the size of the difference between the pre-test and post-test scores relative to the variability in the data. A larger t-value indicates a greater difference between the means. The p-value (< 0.001) indicates the probability of observing such a difference purely by chance. In this case, the p-value is less than 0.001, suggesting that the observed difference is highly unlikely to occur by random chance alone. Therefore, the results of the t-test indicate a statistically significant difference between the pre-test and post-test scores, suggesting that the intervention had a significant impact on the participants' scores.”
- Your manuscript outlines two objectives:
a.To determine if playing a COVID-19 awareness game improved year one nursing students' knowledge about COVID-19.
b.To identify if nursing students were able to identify facts from myths in relation to the transmission of and protective measures required in relation to COVID-19.
However, the results section appears to focus solely on the first objective, analyzing the overall improvement in knowledge scores. The second objective, regarding identifying facts from myths, is not addressed.
Thank you for this comment. We have revised the objectives (from two to one) as this reflects our study and discussion.
Overall, your manuscript has the potential to be a valuable contribution to the field of nursing education. However, before it is ready for publication in a high-impact journal, I recommend addressing the points I specified.
Thank you for your expertise and support.
Reviewer 2 Report
Comments and Suggestions for Authors
Thanks for giving the opportunity to review the article titled “Evaluation of a serious game on nursing student awareness of COVID-19: A pre-test post-test Study.” Thee manuscript is interesting for the reader. However, there are numerous issues that need to be rectified and modified to the manuscript’s quality.
1. The abstract needs to be more precise in accordance with the healthcare MDPI instructions. ( https://www.mdpi.com/journal/healthcare/instructions ). Effectiveness of
2. The study’s aim and the authors’ conclusions are not in a similar context.
3. In general and healthcare journal policy, the abstract must not have a discussion section.
4. Introduction: Globally, even among the general population, the origin, symptoms, vaccines, etc of COVID-19 is well known. Hence, in 2024, it is not necessary to explain all these details with 17 references.
5. The authors started, “It is important to educate undergraduate student nurses about Covid-19 during their 57 training as it may have implications for them in the future.” However, they did not explain it further. The continuity in the introduction is missing. Kindly make a proper conceptualization and link it to the current scenario. Otherwise. The study might be irrelevant to the current (2024) scenario.
6. Also, aims and objectives need to be more refined. Actually, the aim is a long-term (ultimate) one that the authors wanted to achieve. Please rewrite it for better clarification (As the study primarily focuses on short-term knowledge gains immediately following engagement with the serious game).
7. Methods: I strongly suggest following the STROBE checklist while presenting the study. The manuscript is not arranged.
8. Importantly, the ethics statement is confusing. Any study involving human participants requires informed consent. However, the statement mentioned by the authors is not clear.
9. The response rate is low. A response rate of less than 80% always decreases the power.
10. The recruitment plan can be elaborated more.
11. The authors mentioned that the questionnaire was prepared. In which language? What about psychometric properties such as values of validity, reliability, etc?
12. Results: Do the authors collect any background (non-identifiable) characteristics of the study population?
13. Also, the results need more explanation. P-values need to be mentioned in tables.
14. Did the authors make a missing data analysis?
15. The authors did not even attempt to compare their results with other studies. Totally unacceptable in a scientific paper. The first paragraph mainly explains the rationale and objectives again.
16. Why are there two areas in which limitations are discussed?
17. The best part of the manuscript is the conclusion, which gives important insights about rarely explored topics. However, the overall quality of the manuscript sidelined the importance of the wonderful idea.
Comments on the Quality of English LanguageModerately good.
Author Response
- The abstract needs to be more precise in accordance with the healthcare MDPI instructions. ( https://www.mdpi.com/journal/healthcare/instructions ). Effectiveness of
Thank you. We have revised the abstract in accordance with MDPI guidance: Abstract: Background: This study aimed to evaluate the effectiveness of a digital serious game intervention about COVID-19 on pre-registration nursing student knowledge. Method: This study included 282 nursing students from a university in Northern Ireland, with 210 students providing comparable pre-test and post-test results. The 'serious game' aimed to debunk common COVID-19 myths and provide accurate information about the virus. Participants completed a 25-item questionnaire before and after engaging with the game, which included true/false items based on the World Health Organization's list of top COVID-19 myths. The data was analysed using paired t-tests to assess knowledge changes, and scores were calculated as percentages of correct answers. Results: A statistically significant improvement in COVID-19 knowledge was demonstrated among first-year nursing students who engaged with the serious game. The post-test scores (M=92.68, SD=13.59) were notably higher than the pre-test scores (M=82.64, SD=13.26), with a p-value less than 0.001. Conclusion: This research suggests that integrating serious games into undergraduate nursing education can effectively enhance COVID-19 knowledge. This approach is aligned with the evolving trend of employing technology and gamification in healthcare education.
- The study’s aim and the authors’ conclusions are not in a similar context.
Thank you. This was noted by reviewer 1. We have deleted objective two as our paper and discussion focuses solely on the primary objective of knowledge.
- In general and healthcare journal policy, the abstract must not have a discussion section.
Thank you. We have removed the discussion and included a conclusion as per MDPI guidelines.
- Introduction: Globally, even among the general population, the origin, symptoms, vaccines, etc of COVID-19 is well known. Hence, in 2024, it is not necessary to explain all these details with 17 references.
Thank you. We have extensively rewritten the introduction and the references have been removed/changed. All changes to introduction are in red font.
- The authors started, “It is important to educate undergraduate student nurses about Covid-19 during their 57 training as it may have implications for them in the future.” However, they did not explain it further. The continuity in the introduction is missing. Kindly make a proper conceptualization and link it to the current scenario. Otherwise. The study might be irrelevant to the current (2024) scenario.
Thank you. We have deleted this section based on the feedback of the peer-reviewers. We now include the following text about serious games on COVID-19: “Recent studies have highlighted the potential of serious games in addressing COVID-19-related knowledge gaps and promoting behavior change among various populations. For instance, games like Escape COVID-19 and Plague Inc. have been designed to teach infection prevention and control practices, contributing to increased knowledge and attitudes regarding COVID-19 [27-28]. Additionally, serious games like Point of Contact (PoC) have demonstrated success in improving young adults' perceptions of preventive measures, fostering greater awareness and compliance with guidelines [29]. Furthermore, serious games have proven effective in healthcare settings, particularly among frontline workers. Studies have shown that games like Escape COVID-19 have successfully influenced healthcare personnel to adopt COVID-19-safe practices, indicating their potential to enhance IPC behaviours on a national scale [27]. Similarly, serious games like Covidgame have been instrumental in medical education, providing an innovative and engaging platform for students to acquire and retain essential knowledge about COVID-19 [30].”
- Also, aims and objectives need to be more refined. Actually, the aim is a long-term (ultimate) one that the authors wanted to achieve. Please rewrite it for better clarification (As the study primarily focuses on short-term knowledge gains immediately following engagement with the serious game).
The aim and objective (singular) have been rewritten: The aim of this study was therefore to evaluate the effectiveness of a ‘serious game’ around COVID-19 awareness on first year undergraduate nursing students. The objective of this study was to determine if playing a COVID-19 awareness game improved year one nursing students’ knowledge about COVID-19.
- Methods: I strongly suggest following the STROBE checklist while presenting the study. The manuscript is not arranged.
Thank you for this comment. STROBE would not be applicable for this study. However, we have included the following text to present transparency in our methods: “Transparent Reporting of Evaluations with Nonrandomised Designs (TREND) reporting guideline informed the reporting of this intervention evaluation and is available from the corresponding author on reasonable request”.
- Importantly, the ethics statement is confusing. Any study involving human participants requires informed consent. However, the statement mentioned by the authors is not clear.
Thank you. We have modified the text to: Participants were not required to provide written or verbal consent for participation in the study. However, they were explicitly informed that their participation in any of the questionnaires was entirely voluntary. Consent was implied when participants accessed the surveys and chose to complete them.
Further, this approach is very common in research of this type. Therefore, this approach was considered acceptable for several reasons. Firstly, the study involved minimal risk to participants, as it comprised completing questionnaires related to academic topics. Secondly, obtaining written or verbal consent might have introduced bias by discouraging participation or altering participant responses. Additionally, requiring formal consent could have posed logistical challenges, such as ensuring that all participants provided consent before accessing the survey. Therefore, by clearly communicating the voluntary nature of participation and allowing individuals to opt-in by accessing the survey, the study aimed to uphold ethical principles while minimizing barriers to participation. We hope this is satisfactory and our study did receive ethical approval prior to recruitment.
- The response rate is low. A response rate of less than 80% al ways decreases the power.
Thank you for this comment. Given the non-experimental design and type of study, we did not discuss the extent to which a response rate of 68.45% was high or low.
- The recruitment plan can be elaborated more.
Thank you. We have provided more detail: All students, totalling 412, were contacted via email and provided with comprehensive information about the serious game by a person not directly involved in the project. This step ensured that the dissemination of information was impartial and transparent. The provided information sheet outlined the purpose of the research, clarified the consent process, and detailed how participant data would be anonymised to protect privacy and confidentiality. Furthermore, to maintain transparency, the project details and contact information of the research team were readily accessible on the module page where students could access the COVID-19 game.
Students who opted to participate in the evaluation were granted access to the pre- and post-questionnaire links on their individual module homepage. It's important to note that completion of the questionnaires was voluntary; students were not obligated to fill them out to access the game. This approach aimed to uphold the principles of voluntary participation and informed consent while allowing students the flexibility to engage with the game and research activities at their discretion. By providing clear information and accessibility to both the serious game and research materials, the study sought to ensure transparency, respect participants' autonomy, and promote ethical conduct throughout the research process.
- The authors mentioned that the questionnaire was prepared. In which language? What about psychometric properties such as values of validity, reliability, etc?
Thank you. We have reported pilot testing for face validity and updated that the questionnaire was in English. No further psychometric properties were considered due to the nature of the project (e.g., non-funded, non-experimental, evaluation of an innovative serious game).
- Results: Do the authors collect any background (non-identifiable) characteristics of the study population?
No, we did not obtain approval to collect this data because it was not required for our primary objective.
- Also, the results need more explanation. P-values need to be mentioned in tables.
Thank you. We have rewritten the results section with all changes appearing in red font. This has now been addressed.
- Did the authors make a missing data analysis?
No. Given the nature of our research, which was a small-scale, local, non-experimental investigation focusing on students' knowledge and the effectiveness of a serious game, we made a strategic decision to forgo a formal missing data analysis. Our study context involved a homogeneous participant population of nursing students, which reduced the likelihood of significant bias resulting from missing data. Additionally, our primary focus was on descriptive analysis rather than inferential statistics, mitigating the impact of missing data on our study outcomes. Furthermore, the logistical constraints associated with conducting a comprehensive missing data analysis, including limited resources such as time and budget, influenced our decision. We hope this is satisfactory.
- The authors did not even attempt to compare their results with other studies. Totally unacceptable in a scientific paper. The first paragraph mainly explains the rationale and objectives again.
I am disappointed with the tone of this reviewer’s language. It is not supportive to the team that have worked hard to develop this manuscript. That stated, I am hopeful that the manuscript discussion is stronger. We have reduced our objectives (from 1 to 2) as per feedback from the peer-reviewers. Further, we have included a mini synthesis of evidence within our introduction which provides readers with an overview of similar studies in this area. We hope this is satisfactory.
- Why are there two areas in which limitations are discussed?
Thank you. We have included the limitations within one section of the manuscript.
- The best part of the manuscript is the conclusion, which gives important insights about rarely explored topics. However, the overall quality of the manuscript sidelined the importance of the wonderful idea.
Thank you. We appreciate your feedback and believe the revisions have made the manuscript a stronger candidate for publication.
Reviewer 3 Report
Comments and Suggestions for Authors
Dear authors,
This is an exciting study, and the authors are to be congratulated for conducting this work. I recommend several points that need to be considered in your article. I hope it helps you improve the manuscript.
In the introduction section, questionable sources are cited from the outset to represent concepts widely discussed by the WHO, which are preferable to reference (see, for example, citations 1, 2, 3).
Lines 57-64 convey a message lacking linear logical connections. Please review the fluidity of the discourse and the connections between individual statements. Express the concepts in a more discursive manner.
I suggest moving lines 78-80 after lines 73-76 to maintain the coherence of the discourse.
Moreover, the introduction section must explain the rationale and the gap the study intends to address.
In line 162, a citation needs to be included.
Kindly reconsider the sentence structure in lines 171-174.
Evaluate whether Table 1 is necessary for the article and if these data can be incorporated into the text or another table. All figures and tables need more descriptions, are not self-sufficient, and need notes explaining the tables adequately. In particular, Table 2 should present the data differently (range, M(SD), p). Including a table with results for each item of the scale/game would be interesting.
Please revisit the opening sentence of the discussion; starting with lines 228-231 might be more engaging.
Lines 221-224 make a statement that appears to result from the study; please reconsider the sentence structure.
Provide reasoning for the statements extracted from the sentence in lines 224-226.
The aspects covered in paragraphs 237-245 are exciting and relevant to the study. Still, it's surprising that these aspects still need to be considered in the introduction and methodology, even though they have been investigated. Additionally, it's curious why the authors did not delve into the ethical issues related to vaccines (https://doi.org/10.3390/vaccines10101602).
In lines 247-249, some study limitations are presented despite having a dedicated section for limitations. Please consolidate the sections. Furthermore, the number of participants is indicated among the limitations; how can one speak of a limitation if a study on the study's power and minimum sample size has not been conducted?
I hope these suggestions are helpful.
Kind regards.
Comments on the Quality of English LanguageA minor change of the required English language, especially for scientific aspects.
Author Response
This is an exciting study, and the authors are to be congratulated for conducting this work. I recommend several points that need to be considered in your article. I hope it helps you improve the manuscript. In the introduction section, questionable sources are cited from the outset to represent concepts widely discussed by the WHO, which are preferable to reference (see, for example, citations 1, 2, 3).
Thank you. We have revised our introductory paragraph based on feedback from reviewer 1 and have deleted citations 1-7. Our new introduction is highlighted in red font.
Lines 57-64 convey a message lacking linear logical connections. Please review the fluidity of the discourse and the connections between individual statements. Express the concepts in a more discursive manner. I suggest moving lines 78-80 after lines 73-76 to maintain the coherence of the discourse.
Thank you. We have deleted this section and replaced it with a synthesis of the evidence on serious games and Covid: “Recent studies have highlighted the potential of serious games in addressing COVID-19-related knowledge gaps and promoting behaviour change among various populations. For instance, games like Escape COVID-19 and Plague Inc. have been designed to teach infection prevention and control practices, contributing to increased knowledge and attitudes regarding COVID-19 [27-28]. Additionally, serious games like Point of Contact (PoC) have demonstrated success in improving young adults' perceptions of preventive measures, fostering greater awareness and compliance with guidelines [29]. Furthermore, serious games have proven effective in healthcare settings, particularly among frontline workers. Studies have shown that games like Escape COVID-19 have successfully influenced healthcare personnel to adopt COVID-19-safe practices, indicating their potential to enhance IPC behaviours on a national scale [27]. Similarly, serious games like Covidgame have been instrumental in medical education, providing an innovative and engaging platform for students to acquire and retain essential knowledge about COVID-19 [30].”
Moreover, the introduction section must explain the rationale and the gap the study intends to address.
Thank you. We have now included the following paragraph: While serious games show promise in addressing COVID-19 challenges, there are notable limitations [27-30]. These include uncertainties regarding their long-term effectiveness in changing behaviours and the variability in their quality and design. Self-reported measures used in studies may introduce bias, and scalability and accessibility issues remain significant challenges, particularly in low-resource settings. Additionally, the rapid evolution of the pandemic may render some serious games outdated. Addressing these concerns is crucial for maximizing the utility of serious games in combating COVID-19. To our knowledge, there has been no serious game that is co-designed, digital, asynchronous, and designed for solo play. The intervention used in this study focuses on debunking common myths and misconceptions about COVID-19, which remains a problem even after 5 years from the first case of COVID-19, highlighting its importance. Furthermore, its digital and asynchronous format allows for easy updates, ensuring it always reflects the most recent evidence and guidelines”.
In line 162, a citation needs to be included.
Thank you. We have included the reference within the text.
Kindly reconsider the sentence structure in lines 171-174.
Thank you. This has been revised for clarity: The questionnaires were administered to participants before and after they played the serious game. This analysed using paired t-tests to determine if the ‘serious game’ increased nursing students’ knowledge and understanding of COVID-19
Evaluate whether Table 1 is necessary for the article and if these data can be incorporated into the text or another table. All figures and tables need more descriptions, are not self-sufficient, and need notes explaining the tables adequately. In particular, Table 2 should present the data differently (range, M(SD), p). Including a table with results for each item of the scale/game would be interesting.
Thank you. We have kept table 1 in our revision, however have expanded upon table two: Descriptive statistics provided insights into the scores obtained by participants. The mean (M) represents the average score, while the standard deviation (SD) indicates the spread or variability of scores around the mean. In simpler terms, the mean tells us the typical score achieved, while the standard deviation informs us how much the scores deviate from this typical value.
The paired-samples t-test was used to compare the mean scores of the same group of participants before and after an intervention, in this case, the completion of pre-test and post-test questionnaires. It helps determine whether the observed differences between the two sets of scores are statistically significant or merely due to chance.
The t-test result (t(209) = 14.55, p < 0.001) consists of two components: the t-value and the p-value. The t-value (14.55) quantifies the size of the difference between the pre-test and post-test scores relative to the variability in the data. A larger t-value indicates a greater difference between the means. The p-value (< 0.001) indicates the probability of observing such a difference purely by chance. In this case, the p-value is less than 0.001, suggesting that the observed difference is highly unlikely to occur by random chance alone. Therefore, the results of the t-test indicate a statistically significant difference between the pre-test and post-test scores, suggesting that the intervention had a significant impact on the participants' scores. The distribution of pre-test scores (Figure 1) shows that there is an approximately normal distribution with no major positive or negative skew or evidence of ceiling or floor effects.
Please revisit the opening sentence of the discussion; starting with lines 228-231 might be more engaging. Lines 221-224 make a statement that appears to result from the study; please reconsider the sentence structure. Provide reasoning for the statements extracted from the sentence in lines 224-226.
We have rewritten the first two paragraphs of our discussion based on peer-reviewer feedback. This now reads: The pandemic-driven shift to online education has highlighted the critical role of technology-mediated learning, widely acknowledged for its importance in ensuring educational equity and inclusivity [41-43]. Gamification, a strategy with enriched experiential learning, offers students dynamic and innovative avenues to explore various subjects, fostering deeper engagement and understanding [32-36, 44-45]. This study's findings illuminate how pre-registration nursing students' interaction with the COVID-19 serious game enriched their understanding and awareness, empowering them to discern factual information from common misconceptions [46, 47]. Despite the well documented challenges posed by online learning, uptake and participation in this study was high.
Several studies have delved into the efficacy of serious games as an innovative educational tool, demonstrating their potential to enhance understanding of COVID-19 and foster safe infection prevention and control behaviors [27-30]. Building upon this existing body of literature, the present study provided a cohort of first-year undergraduate nursing students with access to a serious game designed to address COVID-19-related knowledge and awareness over a period of thirty days. Incorporating insights from prior research, the study utilised a 25-item questionnaire to assess students' knowledge levels before and after engaging with the COVID-19 serious game. Analysis of pre- and post-test results revealed statistically significant changes in students' knowledge levels following gameplay, aligning with findings from previous COVID-19 COVID-19 serious game studies [27-30, 46-47]. This highlights the potential of serious games as effective educational interventions for enhancing nursing students' understanding of critical public health issues such as COVID-19. Like similar serious game studies, the integration of serious games into nursing education therefore appears to offer an engaging and interactive learning experience and promote active participation and knowledge retention among students [33-34]. By immersing themselves in gameplay, students learners can explore real-world scenarios and apply theoretical knowledge in practical contexts, thereby reinforcing key concepts and skills essential for effective nursing practice
The aspects covered in paragraphs 237-245 are exciting and relevant to the study. Still, it's surprising that these aspects still need to be considered in the introduction and methodology, even though they have been investigated. Additionally, it's curious why the authors did not delve into the ethical issues related to vaccines (https://doi.org/10.3390/vaccines10101602).
Thank you. We have added the final paragraph to provide clarity: Considering the the findings of this study, which demonstrated high engagement and knowledge acquisition through the use of a serious games in the context of COVID-19 education, educators and healthcare institutions may consider leveraging similar gamified approaches to address vaccine hesitancy among nursing students. Serious games could serve as a platform to provide accurate information about COVID-19 vaccines, dispel myths and misconceptions, and address concerns regarding safety and efficacy. Moreover, incorporating interactive elements such as decision-making scenarios related to vaccine administration and public health measures could help nursing students develop the skills and confidence necessary to advocate for vaccination both in clinical practice and within their communities.
In lines 247-249, some study limitations are presented despite having a dedicated section for limitations. Please consolidate the sections. Furthermore, the number of participants is indicated among the limitations; how can one speak of a limitation if a study on the study's power and minimum sample size has not been conducted?
Thank you. We have addressed this and removed the duplicated section.
I hope these suggestions are helpful.
Thank you for your kind and supportive review.
Reviewer 4 Report
Comments and Suggestions for Authors
It seems that the manuscript has already passed some meticulous review stages and as I perceive, the corrections have increased its value significantly.
I have but some minor suggestions:
Double-space to separate sentences (in this case in abstract, but also elsewhere) should not be used with computers.
In row 59, please add references after "Recent studies..."
In row 180 the word "designed" should most probably be substituted with "questionnaire"?
Author Response
Thank you for this supportive feedback.
- Double-space to separate sentences (in this case in abstract, but also elsewhere) should not be used with computers. Thank you. We have done this.
In row 59, please add references after "Recent studies..." Thank you. We have added this.
In row 180 the word "designed" should most probably be substituted with "questionnaire"? Thank you. We have changed this to pre-test, post-test design.
Reviewer 5 Report
Comments and Suggestions for Authors
Abstract: The abstract section details the key aspects of the paper with its purpose, methods, key results and conclusions describe succinctly and in a more scholarly manner.
Keywords: The keywords stated are good and will allow for easy identification of the article when it is finally published in online database searches.
Introduction: The section offers a strong theoretical background to the study. It gives a general overview of the Covid-19 pandemic, its general impacts, its specific impacts on student nurses and the past interventions at using serious games for Covid-19 education. Intelligently, the authors have exposed the research gaps that elevate the relevance of their research by reviewing various serious games that have been used for Covid-19 education, their strengths and limitations. Then at the crucial latter section of the introduction where they lay the brick for their study, they scholarly describe the novelty of their serious game intervention (unique attributes that distinct them from existing and its eventual benefits to the education on Covid-19 myths among student nurses. This is highly commendable.
Materials and Methods: Information on the ethics, description of the intervention, data collection tools and survey instruments administered, and the informed consent on the participation in the evaluation phase of the intervention have been well discussed.
While measures were taken to validate the researcher-designed questionnaire based on the WHO list of Covid-19 myths, information on its analysis, scores that vouch its validity, reliability and consistency have not been provided. Were some revisions made on the questionnaire after the pilot study? What were the Cronbach alpha score, etc.? What scholarly evidence do we have that shows its valid to have been used for the study? Please provide detailed information on this.
Also, information on the study area is missing. Where was the study conducted and what makes the selection of this study area significant for this study?
Results: The results have been well described and presented in much objectivity.
Discussion: The discussion section has intelligently interpreted the study’s results and compared and/or contrasted them with other related studies. What the results suggest in relation to serious games (or the tactful use of gamification) and healthcare education broadly have been exemplified.
Recommendation: The section dwells only on recommendations for future research. While the section is good as its required of any good research, the study must scholarly provide feasible recommendations for practice. This has not been described. How should the revelations of this serious game be deployed generally among student nurses for healthcare education (Covid-19 etc.) and who/which agencies should be tasked with the oversight responsibilities to undertake this exercise?
Strengths and Limitations: This section is good as it describes the strengths and limitations of the study. The latter, a humbling experience that most researchers fail to present for fear it will discredit their work, has been included and explained in a very intelligent manner for readers to appreciate what might have affected the results. The former justified the novelty of the study which is great.
Conclusion: I think the section must clearly state the conclusions from the key results of the study. What does the key results suggest or mean? These conclusions must be stated in much clarity and must be solidly based on the key results obtained.
Generally, it’s a scholarly and innovative study that has been carried out scientifically. Congratulations.
Comments on the Quality of English Language
The quality of the English language is good but the manuscript could benefit from thorough proofreading.
Author Response
Thank you for this supportive review. We have used your feedback to develop the manuscript and note these changes below.
Abstract: The abstract section details the key aspects of the paper with its purpose, methods, key results and conclusions describe succinctly and in a more scholarly manner.
Thank you.
Keywords: The keywords stated are good and will allow for easy identification of the article when it is finally published in online database searches.
Thank you.
Introduction: The section offers a strong theoretical background to the study. It gives a general overview of the Covid-19 pandemic, its general impacts, its specific impacts on student nurses and the past interventions at using serious games for Covid-19 education. Intelligently, the authors have exposed the research gaps that elevate the relevance of their research by reviewing various serious games that have been used for Covid-19 education, their strengths and limitations. Then at the crucial latter section of the introduction where they lay the brick for their study, they scholarly describe the novelty of their serious game intervention (unique attributes that distinct them from existing and its eventual benefits to the education on Covid-19 myths among student nurses. This is highly commendable.
Thank you.
Materials and Methods: Information on the ethics, description of the intervention, data collection tools and survey instruments administered, and the informed consent on the participation in the evaluation phase of the intervention have been well discussed.
Thank you.
While measures were taken to validate the researcher-designed questionnaire based on the WHO list of Covid-19 myths, information on its analysis, scores that vouch its validity, reliability and consistency have not been provided. Were some revisions made on the questionnaire after the pilot study? What were the Cronbach alpha score, etc.? What scholarly evidence do we have that shows its valid to have been used for the study? Please provide detailed information on this.
Thank you for this comment. We have provided the following within the limitations section of the manuscript.
Despite the absence of formal validation procedures, the questionnaire's grounding in authoritative sources such as the WHO, coupled with prior pilot testing, bolsters the confidence in the study's methodology. Future research could explore additional validation methods, such as more extensive psychometric analyses and iterative piloting, to further strengthen the questionnaire's reliability. Nevertheless, for the purposes of this study, the questionnaire's alignment with WHO guidelines provided a solid basis for data collection and analysis.
Also, information on the study area is missing. Where was the study conducted and what makes the selection of this study area significant for this study?
Thank you. We have provided the following information on study setting:
This study took place in one university in Northern Ireland between 04.01.2021 to 28.01.2022. All eligible participants (n=412) were undertaking their first year of a BSc Honours Degree in Professional Nursing
Results: The results have been well described and presented in much objectivity.]
Thank you.
Discussion: The discussion section has intelligently interpreted the study’s results and compared and/or contrasted them with other related studies. What the results suggest in relation to serious games (or the tactful use of gamification) and healthcare education broadly have been exemplified.
Thank you.
Recommendation: The section dwells only on recommendations for future research. While the section is good as its required of any good research, the study must scholarly provide feasible recommendations for practice. This has not been described. How should the revelations of this serious game be deployed generally among student nurses for healthcare education (Covid-19 etc.) and who/which agencies should be tasked with the oversight responsibilities to undertake this exercise?
Thank you for this helpful comment. We have included the following paragraph.
The findings derived from the serious game hold promise for augmenting healthcare education, particularly within nursing. It is recommended that academic institutions consider the integration of serious games into their educational frameworks to complement conventional pedagogical methodologies. Such integration stands to furnish students with immersive, experiential learning modalities, thereby enriching their comprehension of pivotal healthcare tenets, including the management of COVID-19. Further, given the critical importance of vaccine acceptance in combating infectious diseases such as COVID-19, efforts could also be directed towards leveraging serious games as brief interventions to address myths associated with vaccine hesitancy among healthcare professional students and healthcare workers.
Strengths and Limitations: This section is good as it describes the strengths and limitations of the study. The latter, a humbling experience that most researchers fail to present for fear it will discredit their work, has been included and explained in a very intelligent manner for readers to appreciate what might have affected the results. The former justified the novelty of the study which is great.
Thank you.
Conclusion: I think the section must clearly state the conclusions from the key results of the study. What does the key results suggest or mean? These conclusions must be stated in much clarity and must be solidly based on the key results obtained.
Thank you. We have included the following paragraph within our conclusion.
Through the implementation of a pre-test and post-test design, significant improvements in participants' knowledge levels were observed following engagement with the COVID-19 serious game. These findings highlight the potential of serious games as innovative educational interventions for promoting active learning and fostering essential skills requisite for effective nursing practice. The study's recommendations advocate for the integration of serious games into nursing curricula to complement traditional pedagogical approaches, providing students with immersive, experiential learning modalities. Furthermore, given the imperative of vaccine acceptance in combating infectious diseases like COVID-19, serious games may offer a promising avenue for addressing vaccine hesitancy among healthcare professional students and workers.
Generally, it’s a scholarly and innovative study that has been carried out scientifically. Congratulations.
Thank you for this supportive and helpful review.
Round 2
Reviewer 3 Report
Comments and Suggestions for Authors
Thank you for making the necessary revisions to improve this manuscript.
Author Response
Thank you for this supportive review.